# Gas Chromatography-Mass Spectrometry (GC-MS) Analysis of Essential Oils from AgNPs and AuNPs Elicited *Lavandula angustifolia* In Vitro Cultures

**DOI:** 10.3390/molecules24030606

**Published:** 2019-02-09

**Authors:** Aneta Wesołowska, Paula Jadczak, Danuta Kulpa, Włodzimierz Przewodowski

**Affiliations:** 1Faculty of Chemical Technology and Engineering, Department of Organic and Physical Chemistry, West Pomeranian University of Technology in Szczecin, Aleja Piastów 42, 71-065 Szczecin, Poland; aneta.wesolowska@zut.edu.pl; 2Department of Plant Genetics, Breeding and Biotechnology, Faculty of Environmental Management and Agriculture, West Pomeranian University of Technology in Szczecin, ul. Słowackiego 17, 71-434 Szczecin, Poland; paula.jadczak@zut.edu.pl; 3Plant Breeding and Acclimation Institute—National Research Institute, Bonin Reaserch Centre, Bonin 3, 73-009 Bonin, Poland; wlodzimierz.przewodowski@ihar.edu.pl

**Keywords:** nanoparticles, secondary metabolites, shoot cultures, micropropagation, elicitor

## Abstract

The aim of this study was to determine how the addition of gold and silver nanoparticles to culture media affects the composition of essential oils extracted from *Lavandula angustifolia* propagated on MS media with the addition of 10 and 50 mg·dm^−3^ of gold (24.2 ± 2.4 nm) and silver (27.5 ± 4.8 nm) nanocolloids. The oil extracted from the lavender tissues propagated on the medium with 10 mg·dm^−3^ AgNPs (silver nanoparticles) differed the most with respect to the control; oil-10 compounds were not found at all, and 13 others were detected which were not present in the control oil. The addition of AuNPs (gold nanoparticles) and AgNPs to the media resulted in a decrease of lower molecular weight compounds (e.g., α- and β-pinene, camphene, δ-3-carene, p-cymene, 1,8-cineole, trans-pinocarveol, camphoriborneol), which were replaced by those of a higher molecular weight (τ- and α-cadinol 9-cedranone, cadalene, α-bisabolol, cis-14-nor-muurol-5-en-4-one, (*E*,*E*)-farnesol).

## 1. Introduction

Nanotechnology has become one of the fastest growing interdisciplinary fields of science today. Nanoparticles, i.e., compounds or elements reduced to the size of less than 100 nanometers, differ in terms of their atomic structure compared to the material they are derived from, and also differ in terms of their physical, chemical, and biological properties. The most important advantage of nanoparticles is a high surface-to-volume ratio, which tends to increase with the reduction in their diameter, whereby nanoparticles demonstrate very high chemical activity [1]. Their highly developed active surface area significantly affects their adsorption properties, material reactivity, and antimicrobial properties [2].

The most widely used and known nanoparticles are those of precious metals: Gold and silver. They undergo various processes which are not observed in macroscopic environments. Nanosilver has antimicrobial (antifungal and antibacterial) properties. Gold in its nanoform offers therapeutic effects due to its ease to penetrate body cells where it strongly stimulates their regeneration [3,4]. Nanoparticles of precious metals form stable colloidal solutions, which can be applied to plant in vitro cultures [5].

Nanoparticles are easily absorbed and accumulated by plants. The processes of nanopenetration into the cells of living organisms are still to be explored in detail. However, it has been confirmed they enter certain cells through endocytosis or through pass through surface pores in plant cell walls [6,7]. The selective properties of cell walls enable the transport of particles measuring from 5 to 20 nm, allowing nanoparticles to easily penetrate cells and spread throughout the entire organism, ultimately affecting biological processes occurring in the cells [8].

With their unique nanostructural properties, these materials are used in many key industries, such as pharmaceuticals, electronics, cosmetology, medicine, environmental protection, textiles, and packaging. They are also applied in biotechnology, and recently, in plant in vitro cultures [9]. Nanosilver is used in plant in vitro cultures at the culture initiation stage to prevent contaminations, offering a viable alternative to antibiotics used in plant micropropagation [10].

Ongoing studies are attempting to determine the suitability of nanoparticles as elicitors in the in vitro cultures. Currently used elicitors are either biotic agents derived from biological sources, such as components of fungal and bacterial cell wall structures (polysaccharides, glycoproteins, inactivated enzymes, curdlan, chitosan), or abiotic factors of chemical or physical origin (heavy metal salts, osmotic stress, mechanical damage, ultraviolet radiation) [11]. Nanometal particles have shown a high capacity for attaching to plant tissues and activate enzymatic pathways responsible for the production of secondary metabolites [12]. They also contribute to the peroxidation of cellular membranes in plant cells and influence the expression of genes responsible for the production of biologically active compounds [13].

So far, attempts to use gold and silver nanocolloids as elicitors in plant in vitro cultures have been limited. The addition of these substances increased the production of secondary metabolites in the cultures of *Salvia miltiorrhiza* [14], *Artemisia annua* [15], *Brugmensia candida* [16], *Corylus avellana* [17], *Prunella vulgaris* [18], and *Aloe vera* [13]. The influence of nanoparticles on plants depends on several factors, such as plant species, its age, growing conditions, culture medium, exposure time of the plant to nanomaterial, and administration method.

Essential oils constitute mixtures of volatile compounds, sesquiterpenes, and primarily monoterpenes [19]. The main components of the essential oils isolated from *L. angustifolia* tissue are, among others, linalool, borneol, geraniol, and linalool acetate [20,21]. The composition of an essential oil depends mainly on the plant genotype, yet its composition may differ under the influence of developmental and environmental factors, i.e., sun exposure, plant age, seedling collection method or essential oil isolation method [22,23]. 

There have been no literature reports so far regarding the influence of nanoparticles on the production of essential oils by plants propagated in in vitro cultures. The studies by Hatami et al. [24] and Ghanati and Bakhtarian [25] show that the application of metal nanoparticles to plants growing under natural conditions results in a change in the essential oil content extracted from their tissues. The aim of this study was to verify how gold and silver nanocolloids influence the composition of essential oils in narrow-leaved lavender (*Lavandula angustifolia*) propagated in in vitro cultures.

## 2. Results and Discussion

Hydrodistillation of the dried leafy shoots of *Lavandula angustifolia* generated pale yellow liquids with a yield from 0.81% (10 mg·dm^−3^ AuNPs) to 1.27% (10 Ag mg·dm^−3^ NPs) (Table 1). Interestingly, the results of this study are comparable with those obtained from lavender flowers. Kara and Baydar [26] studied four lavender cultivars and indicated that the oil content varied from 0.35 to 2.0%. Zheljazkov et al. [27] reported the content of volatile oil in dried flowers to range from 0.71 to 1.30%. However, the content of volatile oil in the leaves of lavender cultivated in Northwest Iran (0.64%) was lower when compared with the results discussed herein [28].

The chemical composition of *L. angustifolia* essential oils is shown in Table 2 and Table 3, where the percentage composition and retention indices of the constituents are given. A total of 97 different compounds representing 99.29–99.95% of the oils were identified. The main volatile constituents were: Borneol (12.14–16.46%), τ-cadinol (12.96–16.63%), caryophyllene oxide (8.79–12.23%), γ-cadinene (4.54–6.08%), and 1,8-cineole (2.80–4.58%). Other important constituents were: Cis-14-nor-muurol-5-en-4-one (2.68–4.45%), β-pinene (1.93–3.14%), camphor (2.05–2.79%), and α-santalene (1.42–2.64%). The extracted oils were the most abundant in oxygenated sesquiterpenes (36.34–43.36%), followed by oxygenated monoterpenes (27.77–38.23%), sesquiterpene hydrocarbons (10.27–14.35%), and monoterpene hydrocarbons (5.57–10.40%).

The growing medium applied affected the percentage composition of each essential oil constituent. The highest concentrations of borneol (16.46%) and 1,8-cineole (4.58%) were noticed in the volatile oil isolated from plants cultivated on the medium supplemented with gold nanoparticles (50 mg·dm^−3^ AuNPs). Addition of silver nanoparticles (50 mg·dm^−3^ AgNPs) to the growing medium increased the content of γ-cadinene (6.08%) and caryophyllene oxide (12.23%) in the oil (Figure 1). 

However, the percentage content of camphor was lower in the plants cultivated on the medium supplemented with AgNPs (10 and 50 mg·dm^−3^). Moreover, volatile oil derived from lavender cultivated on MS medium was richer in β-pinene (3.14%), α-pinene (1.46%), *p*-cymene (1.39%), camphene (1.08%), and δ-3-carene (0.93%).

Phytochemical studies revealed that linalool (9.3–68.8%) and linalyl acetates (1.2–59.4%) were the main components of the aerial parts and flowers of *Lavandula angustifolia* [29,30]. However, the essential oil obtained from plants cultivated in North Africa [31] had 1,8-cineole (29.4%) and camphor (24.6%) as the major constituents. 1,8-cineole (65.4%) and borneol (11.5%) dominated in the essential oils isolated from the leaves of *L. angustifolia* collected near Isfahan, Iran [32]. Borneol was the main compound in the essential oils isolated from leafy stems of three lavender cultivars propagated in in vitro cultures: ‘Blue River’ (25.75%), ‘Elegance Purple’ (32.17%), and Munstead (13.38%) [33].

The percentage contents of linalool (0.30–0.77%), 1,8-cineole (2.80–4.58%), and camphor (2.05–2.79%) found in volatile oils in this study were much lower than the results reported in the referenced literature. Essential oils isolated from plants grown on control medium and 50 mg·dm^−3^ AuNPs medium were the only ones with higher borneol content (16.00–16.46%) compared with the results obtained by Andrys and Kulpa [29]. Linalool, lavandulol, and their esters (linalyl acetate and lavandulyl acetate) are responsible for the fresh and floral smell of lavender oil. Moreover, the quality of oil depends on both a high content of linalool and linalyl acetate and their mutual proportions (preferably higher than 1) [34].

Contrary to the results obtained by other researches, lavandulol and linalyl acetate were not detected in the oils in this study, and the content of lavandulyl acetate did not exceed 0.21%. The data reported in the literature indicated that many terpenoids are biologically active and are used medicinally [35]. Camphor, with its specific camphoraceous odor, is used commercially as a moth repellent and as a preservative in pharmaceuticals and cosmetics [36]. Borneol, a widely-used food and cosmetic additive, possesses analgesic, anti-inflammatory, and antibacterial properties [37,38]. It is well known that 1,8-cineole and camphor are responsible for the insecticidal activity of the plants from *Lavandula* genus [39]. Based on these facts, it can be stated that the volatile oils extracted from leafy shoots of *L. angustifolia* may have commercial applications.

The oil extracted from the tissues of lavender propagated on the culture medium which was supplemented with 10 mg·dm^−3^ of AgNPs differed the most with respect to the control culture (plants propagated on the culture medium with no nanoparticles) in terms of the number of compounds: While 10 compounds were not found in it at all, 13 others were detected which were not observed in the control oil. The addition of AuNPs and AgNPs to the media resulted in a decrease in compounds with lower molecular weight (e.g., α- and β-pinene, camphene, δ-3-carene, p-cymene, 1,8-cineole (eucalyptol), trans-pinocarveol, camphor, and borneol), which were replaced by those of higher molecular weight (τ- and α-cadinol 9-cedranone, cadalene, α-bisabolol, cis-14-nor-Muurol-5-en-4-one, (*E*,*E*)-farnesol).

Heavy metal nanocolloids that have been recently used in plant in vitro cultures, as elicitors provoke the production of secondary metabolites. There are research reports confirming that these particles are capable of eliciting responses in plants consistent with those generated when typical elicitors are used [40,41]. It is commonly believed that the production of secondary metabolites in plants is significantly affected by environmental stress. Biotic and abiotic stresses delay cellular differentiation through the production of reactive oxygen species (ROS), which directly destroy cells by producing secondary metabolites [42,43]. The researchers suggest that oxidative stress induced by nanoparticles is correlated with the production of secondary metabolites in plants. Due to their small size, nanocolloids can easily attach to plant cell walls, destroy them, change their permeability, and thus significantly affect cellular metabolism [13]. Zhang et al. [14] confirmed the effectiveness of silver as an elicitor using silver ions in the production of diterpenoids in the cultures of root hairs of *Salvia miltiorrhiza* genus. The addition of silver to the culture media of root hairs resulted in an increase in the production of reactive oxygen species. Activation of ROS-based mechanisms following exposure of plants from *Calendula officinalis* L. genus to nanoparticles was also confirmed by Ghanati and Bakhtiarian [20] in the production of secondary metabolites. Fazal et al. [18] demonstrated that a callus of *Prunella vulgaris* genus treated with silver and gold nanocolloids produced significant quantities of antioxidant enzymes, such as POD and SOD, as well as phenolic and flavonoid compounds that are directly related to the protection of plants against environmental stress. Silver nanocolloids were used to produce capsaicin from *Capsicum* sp. and resulted in a significant increase in the production of this compound [44]. Hemm et al. [45] and Liu et al. [46] showed that growth regulators combined with elicitors resulted in a larger organogenic potential of plants and increased the production of primary and secondary metabolites. 

The study showed that the addition of gold and silver nanocolloids to the culture media significantly affected the composition of essential oil derived from narrow-leaved lavender cultivated in in vitro cultures. In the oils extracted from plants propagated in vitro on culture media with the addition of nanoparticles, a variety of compounds were identified that were not present in the oil derived from plants grown on the control medium. The above suggests that gold and silver nanoparticles can be successfully used to obtain essential oils of different composition which may result in different properties: Fragrance and, above all, antioxidant and antimicrobial activity, but the latter requires further studies. It is also necessary to determine the toxicity of nanoparticles in relation to plant tissues.

## 3. Material and Methods

### 3.1. Nanoparticles

Aqueous suspensions of gold and silver nanoparticles were synthesized using Turkevich et al. [47] and Liu et al.’s [48] methods with modified synthesis conditions and a two-stage microwave-convection heating method. For this purpose, aqueous mixtures of 0.903 g·dm^−3^ of sodium citrate with 2.378 g·dm^−3^ of tetrachloroauric acid (HAuCl_4_), and 1.189 g·dm^−3^ of silver nitrate (AgNO_3_), respectively, were prepared. After their purification with a small (0.2 μm) pore antibacterial filter (Sartorius, Goettingen, Germany), they were placed in a microwave (MX 245), where they were stirred and heated to 100 °C at 800 W, which allowed for reaching a heating rate of approx. 1.6 °C/s. Once the preset temperature was reached, the mixture was kept in the microwave for an additional 20 s, and then placed in HBR 4 digital IKAMAG heating bath (IKA, Staufen, Germany), where it was stirred with a magnetic stirrer and incubated at 95 °C for an additional 15 min. It was then gradually cooled to room temperature (0.8 °C/min). To obtain a similar distribution of nanoparticle diameters, the resulting mixtures were homogenized in a centrifugal force field in Beckmann JA-20 centrifuge to obtain similar nanoparticle concentrations in both mixtures. Once their spectra were plotted with UV-VIS EPOCH microplate spectrophotometer (BioTek, Bad Friedrichshall, Germany), the optical density of the fractions obtained was adjusted to a common DEV value, using the following spectra absorbance maxima λ_max_ = 520 nm and λ_max_ = 445 nm for gold and silver colloids, respectively (Figure 2). The similarities in morphology, shape, and size of the synthesized and prepared gold (24.2 ± 2.4 nm) and silver (27.5 ± 4.8 nm) nanoparticles were assessed after their application to the surface of a nylon membrane (Supelco, Park Bellefonte, PA, USA) and through an analysis of images from a scanning electron microscope (SEM, FEI Quanta 200 FEG model) (FEI Company, Tokyo, Japan, Figure 3).

### 3.2. In Vitro Cultures

The materials examined in this study were plants of narrow-leaved lavender (*Lavendula angustifolia*), ‘Munstead’ cultivar. Single-node shoot fragments with a length of 1–1.5 cm were put in glass jars with a capacity of 300 mL, filled with 30 mL of the medium. The media, with a mineral composition developed by Murashige and Skoog [49] (MS media), were supplemented with 2 mg·dm^−3^ kinetin (KIN) and 0.2 mg·dm^−3^ indole-3-acetic acid (IAA) [50] with the addition of gold (AuNPs) with a diameter of 24.2 ± 2.4 nm and silver (AgNPs) with a diameter of 27.5 ± 4.8 nm nanocolloids with the concentrations of: 10 and 50 mg·dm^−3^, respectively. Furthermore, the media were supplemented with: 30 g·dm^−3^ of sucrose, 100 mg·dm^−3^ of inositol, and solidified with agar at 7 g·dm^−3^. Medium pH was set at 5.7 using 0.1 M solutions of HCl and NaOH. The jars were sterilized at 121 °C for 20 min. The jars with cultures were placed in a phytotron, with a humidity of 70–80% and temperature of 24 °C. The cultures were illuminated for 16 h a day, and the illuminance was kept at 35 µEM^−2^s^−1^ PAR.

### 3.3. Extraction of Essential Oils

Fifteen grams of the entire dried aerial parts of lavender were placed in 1000 mL round-bottomed flasks along with 400 mL of distilled water and subjected to hydrodistillation (3 replicates) for two hours using a Clevenger apparatus as recommended by the European Pharmacopoeia 5.0 [51]. The essential oil extracts were dried over anhydrous sodium sulfate, filtered, weighed and stored in dark sealed vials at 4 °C until gas chromatography/mass spectrometry (GC-MS) analysis was performed. Essential oil percentage was calculated based on the dry weight of plant material and expressed as (% *w*/*w*) in Table 1.

### 3.4. Gas Chromatography/Mass Spectrometry (GC-MS) Analyses of Essential Oils

The qualitative GC-MS analysis of the extracted essential oils was carried out using HP 6890 gas chromatograph coupled with HP 5973 Mass Selective Detector (Agilent Technologies, Foster City, CA, USA) operating in 70 eV mode. Samples of 2 μL (40 mg of oil dissolved in 1.5 mL of dichloromethane) were injected in a split mode at a ratio of 5:1. The compounds were separated on a 30 m long capillary column (HP-5MS), 0.25 mm in diameter and with 0.25 µm thick stationary phase film ((5% phenyl)-methylpolysiloxane).

The flow rate of helium through the column was kept at 1.2 mL·min^−1^. The initial temperature of the column was 45 °C, then it was increased to 200 °C at a rate of 5 °C·min^−1^ (kept constant for 10 min), and then heated up to a final temperature of 250 °C at a rate of 5 °C min^−1^. The oven was kept at this temperature for 20 min. The injector temperature was 250 °C, the transfer line temperature was 280 °C, and the ion source temperature was 230 °C. The solvent delay was 4 min. The scan range of the MSD was set at 40 to 550 m*m*/*z*. The total running time for a sample was about 71 min. The relative percentage of the essential oil constituents was evaluated from the total peak area (TIC) by apparatus software [52,53]. Essential oil constituents were identified by comparison of their mass spectra with those stored in the Wiley NBS75K.L and NIST/EPA/NIH (2002 version, National Institute of Standards and Technology, Gaithersburg, MD, USA) mass spectral libraries using various search engines (PBM, Nist02). The identity of compounds was also confirmed by comparison of their calculated retention indices with those reported in NIST Chemistry WebBook (http://webbook.nist.gov/chemistry/). For retention indices (RI) calculation [54,55], a mixture of homologus series of n-alkanes C_7_-C_40_ (Supelco, Bellefonte, PA, USA) was used, under the same chromatographic conditions which were applied for the analysis of the lavender essential oils.

## Figures and Tables

**Figure 1 molecules-24-00606-f001:**
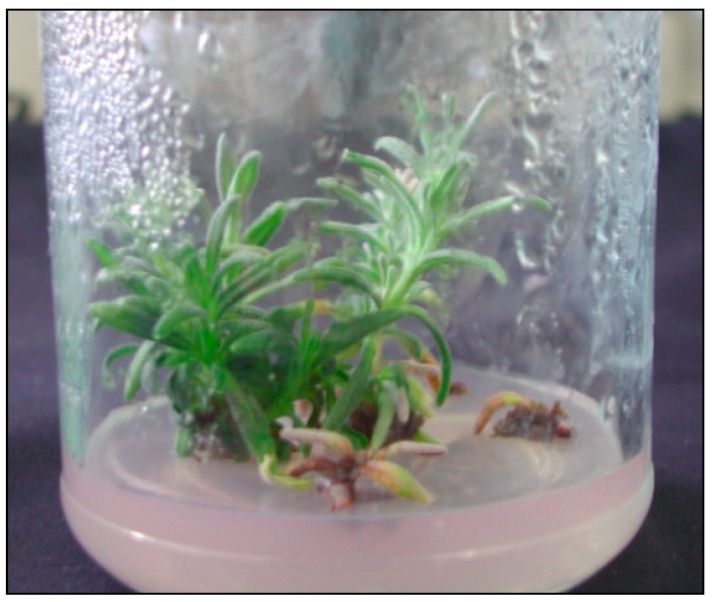
Plants of *Lavandula angustifolia* Mill. propagated on medium with 50 mg·dm^−3^ AuNPs.

**Figure 2 molecules-24-00606-f002:**
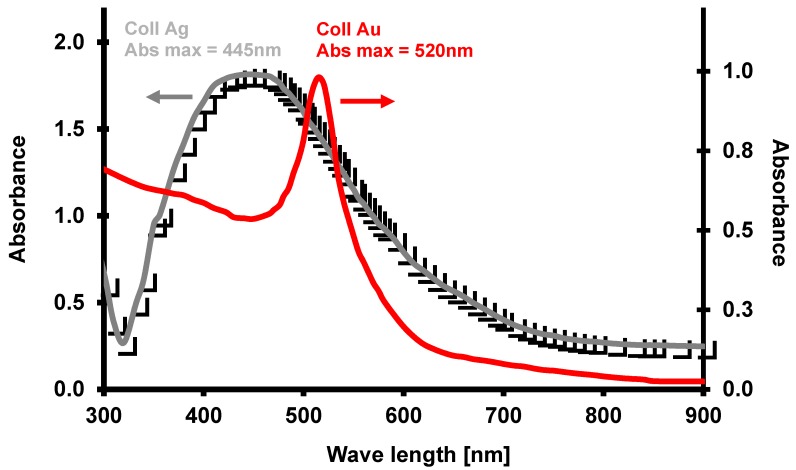
UV-VIS spectral spectra of the fraction of 4-5000 x g of gold and colloidal silver.

**Figure 3 molecules-24-00606-f003:**
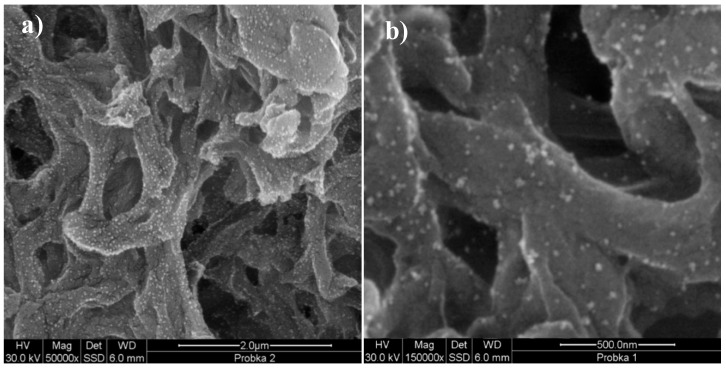
View of nanoparticles of gold colloids (**a**) and silver (**b**) after fractionation and placement on a nylon diaphragm (Supelco, Park Bellefonte, PA, USA) made with a help of FEI Quanta 200 FEG scanning electron microscope. The magnification applied to the observed colloidal gold and silver 50,000 (**a**) and 150,000 (**b**) times, respectively.

**Table 1 molecules-24-00606-t001:** Essential oil content in lavender plants as a function of medium used.

Medium [mg·dm^−3^]	Essential Oil Content (%*w*/*w*)
0—control	1.15
50 Au	0.95
10 Au	0.81
50 Ag	0.82
10 Ag	1.27

**Table 2 molecules-24-00606-t002:** Statistical analysis of main compounds.

Compound	RI	Control	50 mg·dm^−3^ Au	10 mg·dm^−3^ Au	50 mg·dm^−3^ Ag	10 mg·dm^−3^ Ag
**1**	α-Pinene	933	1.46a	0.99b	1.03b	1.11b	0.64c
2	β-Pinene	977	3.14a	2.56b	2.25b	2.53b	1.93b
3	p-Cymene	1025	1.39a	1.18b	1.17b	0.91c	0.92c
4	1,8-Cineole	1031	4.49a	4.58a	2.80b	2.95b	2.95b
5	*trans*-Pinocarveol	1140	1.61a	1.65a	1.54b	1.42c	1.14d
6	Camphor	1145	2.75a	2.79a	2.41b	2.06c	2.05c
*7*	Pinocarvone	1164	1.32a	1.36a	1.32a	1.16b	0.90c
8	Borneol	1170	16.00a	16.46a	12.78b	12.14b	12.99b
9	Myrtenol	1198	2.25a	2.35a	1.94a	2.13a	1.85a
10	Geranylacetate	1385	1.20a	1.38a	1.14a	0.59b	1.41a
11	α-Santalene	1422	1.90bc	1.42d	2.16b	2.64a	1.74c
12	γ-Cadinene	1518	4.97c	4.54d	5.09c	6.08a	5.36b
13	Caryophylleneoxide	1589	9.12c	8.54c	11.06b	12.23a	8.79c
14	τ-Cadinol	1648	12.96c	14.35b	13.65bc	14.17b	16.63a
16	α-Cadinol	1662	1.33a	1.13b	1.35a	1.36a	1.08c
17	Cadalene	1675	1.87d	1.58e	2.34b	2.53a	2.03c
18	*cis*-14-nor-Muurol-5-en-4-one	1693	2.68c	3.72b	3.72b	3.37c	4.45a
19	(*E*,*E*)-Farnesol	1720	1.20d	1.43b	1.32c	1.45b	1.57a
20	Bisabolol oxide A	1750	1.97c	2.26b	2.23b	2.13b	2.60a

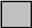
 compounds with a significantly lower content as compared with the control oil; 
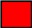
 compounds with a significantly higher content as compared with the control oil; a, b, c—values followed by the same letter are not significantly different at *p* ≤ 0.05 according to the LSD (least significant differences) Tukey test.

**Table 3 molecules-24-00606-t003:** Relative percentage composition of lavender essential oils depending on the medium ±SD (*n* = 3).

No.		Compound	RI	Control		50 mg·dm^−3^ Au		10 mg·dm^−3^ Au		50 mg·dm^−3^ Ag		10 mg·dm^−3^ Ag	
1.	MH	α-Thujene	927	0.09	±0.01								
2.	MH	α-Pinene	933	1.46	±0.16	0.99	±0.10	1.03	±0.01	1.11	±0.18	0.64	±0.04
3.	MH	Camphene	948	1.08	±0.16	0.78	±0.07	0.85	±0.04	0.77	±0.11	0.48	±0.03
4.	MH	Thuja-2,4(10)-diene	954			0.09	±0.01						
5.	MH	β-Thujene	971	0.33	±0.02	0.24	±0.01	0.28	±0.01	0.20	±0.00	0.21	±0.01
6.	MH	Sabinene	974	0.31	±0.01	0.25	±0.03	0.23	±0.01	0.23	±0.03	0.21	±0.01
7.	MH	β-Pinene	977	3.14	±0.26	2.56	±0.26	2.25	±0.08	2.53	±0.46	1.93	±0.09
8.	MH	δ-3-Carene	1010	0.93	±0.08	0.72	±0.06	0.77	±0.03	0.78	±0.10	0.59	±0.03
9.	MH	m-Cymene	1022	0.52	±0.05	0.45	±0.04	0.49	±0.02	0.33	±0.01	0.32b	±0.01
10.	MH	p-Cymene	1025	1.39	±0.13	1.18	±0.10	1.17	±0.07	0.91	±0.06	0.92c	±0.04
11.	MH	D-Limonene	1029	0.81	±0.11	0.33	±0.47	0.64	±0.02	0.58	±0.04	0.27a	±0.38
12.	OM	1,8-Cineole	1031	4.49	±0.26	4.58	±0.01	2.80	±0.12	2.95	±0.43	2.95b	±0.33
13.	MH	γ-Terpinene	1060	0.10	±0.01	0.08	±0.01						
14.	OM	*cis*-Sabinenehydrate	1068	0.070	±0.00							0.09	±0.00
15.	MH	α-Terpinolene	1091	0.24	±0.01	0.23	±0.00	0.27	±0.02	0.15	±0.06		
16.	OM	Linalool	1101	0.65	±0.03	0.77	±0.05	0.55	±0.04	0.30	±0.06	0.51b	±0.01
17.	O	α-Pineneoxide	1110	0.210	±0.01								
18.	OM	Fenchol	1114			0.21	±0.01	0.26	±0.01	0.16	±0.01	0.19	±0.00
19.	O	3-Octanol acetate	1122	0.15	±0.00	0.15	±0.01	0.20	±0.01	0.13	±0.01		
20.	OM	α-Campholenal	1127	0.23	±0.00	0.22	±0.01	0.24	±0.01	0.21	±0.01	0.20	±0.01
21.	OM	1,2-Dihydrolinalool	1136	0.24	±0.01	0.21	±0.01	0.26	±0.01	0.20	±0.01	0.18	±0.00
22.	OM	*trans*-Pinocarveol	1140	1.61	±0.04	1.65	±0.10	1.54	±0.08	1.42	±0.05	1.14	±0.02
23.	OM	Camphor	1145	2.75	±0.08	2.79	±0.21	2.41	±0.13	2.06	±0.13	2.05	±0.03
24.	OM	Pinocarvone	1164	1.32	±0.01	1.36	±0.08	1.32	±0.08	1.16	±0.01	0.90	±0.00
25.	OM	Borneol	1170	16.00	±0.58	16.46	±1.51	12.78	±0.44	12.14	±1.20	12.99	±0.16
26.	OM	Terpinen-4-ol	1179	0.69	±0.04	0.64	±0.04	0.62	±0.01	0.60	±0.06	0.48	±0.00
27.	OM	p-Cymen-8-ol	1184	0.80	±0.04	0.84	±0.01	0.99	±0.06	0.57	±0.07	0.40	±0.02
28.	OM	Cryptone	1187	0.55	±0.25	0.71	±0.03	0.77	±0.04	0.52	±0.06	0.35	±0.01
29.	OM	α-Terpineol	1193	0.66	±0.01	0.65	±0.04	0.51	±0.00	0.52	±0.06	0.41	±0.01
30.	OM	Myrtenol	1198	2.25	±0.06	2.35	±0.11	1.94	±0.06	2.13	±0.22	1.85	±0.03
31.	OM	Verbenone	1210	0.78	±0.06	0.69	±0.08	0.61	±0.02	0.51	±0.07	0.45	±0.01
32.	OM	*cis*-Carveol	1221	0.18	±0.01	0.18	±0.00	0.24	±0.01	0.08	±0.11		
33.	OM	*trans*-Carveol	1224	0.20	±0.00	0.23	±0.01	0.23	±0.00	0.09	±0.12		
34.	OM	Bornylformate	1229	0.86	±0.01	0.85	±0.04	0.63	±0.00	0.62	±0.02	0.91	±0.01
35.	OM	d-Carvone	1247	0.24	±0.00	0.24	±0.01	0.32	±0.01	0.10	±0.14	0.20	±0.02
36.	OM	Geraniol	1254	0.46	±0.01	0.43	±0.01	0.40	±0.01	0.33	±0.02	0.31	±0.01
37.	OM	α-Citral	1272	0.12	±0.01	0.12	±0.00	0.11	±0.01				
38.	OM	Bornylacetate	1287	0.32	±0.01	0.29	±0.00	0.32	±0.04	0.27	±0.00	0.38	±0.03
39.	OM	Lavandulylacetate	1292	0.20	±0.00	0.16	±0.00	0.21	±0.01	0.19	±0.01	0.18	±0.01
40.	OM	Piperitenone	1341	0.11	±0.01	0.12	±0.01	0.14	±0.00	0.05	±0.06	0.12	±0.01
41.	OM	Nerylacetate	1367			0.10	±0.01						
42.	OM	Geranylacetate	1385	1.20	±0.06	1.38	±0.07	1.14	±0.06	0.59	±0.28	1.41	±0.27
43.	SH	α-Cedrene	1416	0.40	±0.06	0.39	±0.04	0.45	±0.04	0.47	±0.01	0.45	±0.04
44.	SH	α-Santalene	1422	1.90	±0.04	1.42	±0.02	2.16	±0.01	2.64	±0.00	1.74	±0.04
45.	SH	α-Bergamotene	1438	0.28	±0.01	0.25	±0.01	0.33	±0.01	0.38	±0.02	0.29	±0.00
46.	SH	Aromadendrene	1448	0.11	±0.01	0.08	±0.00	0.14	±0.00	0.14	±0.01	0.12	±0.01
47.	SH	β-Santalene	1450	0.11	±0.01			0.12	±0.01	0.15	±0.01	0.10	±0.00
48.	SH	*trans*-β-Bergamotene	1460	0.15	±0.01	0.09	±0.04			0.16	±0.06		
49.	SH	β-Chamigrene	1463	0.11	±0.01	0.09	±0.00	0.11	±0.00	0.14	±0.01	0.12	±0.00
50.	SH	Di-epi-α-Cedrene	1470	0.13	±0.00	0.14	±0.01	0.15	±0.00	0.15	±0.01	0.15	±0.01
51.	SH	*cis*-β-Farnesene	1488	0.10	±0.00	0.12	±0.06	0.19	±0.05	0.15	±0.04	0.16	±0.01
52.	SH	β-Bisabolene	1511							0.07	±0.09		
53.	SH	γ-Cadinene	1518	4.97	±0.06	4.54	±0.11	5.09	±0.03	6.08	±0.03	5.36	±0.05
54.	SH	β-Sesquiphellandrene	1522	0.39	±0.01	0.44	±0.04	0.56	±0.01	0.50	±0.01	0.52	±0.01
55.	SH	δ-Cadinene	1526	0.45	±0.01	0.42	±0.03	0.49	±0.01	0.54	±0.01	0.50	±0.02
56.	SH	*trans*-Calamenene	1533	0.24	±0.00	0.24	±0.03	0.33	±0.01	0.32	±0.03	0.35	±0.01
57.	SH	Cadina-1,4-diene	1536	0.53	±0.01	0.60	±0.06	0.59	±0.03	0.53	±0.02	0.60	±0.01
58.	SH	α-Cadinene	1543							0.19	±0.01	0.13	±0.18
59.	SH	α-Calacorene	1547	0.36	±0.01	0.52	±0.06	0.51	±0.03	0.45	±0.05	0.48	±0.01
60.	SH	Germacrene B	1557	0.89	±0.02	0.82	±0.07	1.14	±0.03	1.23	±0.05	0.86	±0.01
61.	SH	β-Calacorene	1563	0.10	±0.01	0.11	±0.03			0.06	±0.08	0.13	±0.01
62.	OS	Nerolidol	1569	0.51	±0.01	0.64	±0.07	0.62	±0.02	0.60	±0.02	0.66	±0.01
63.	O	(*Z*)-3-Hexenyl benzoate	1579	0.63	±0.01	0.76	±0.08	0.74	±0.04	0.73	±0.01	0.76	±0.01
64.	OS	Caryophylleneoxide	1589	9.12	±0.16	8.54	±0.31	11.06	±0.13	12.23	±0.21	8.79	±0.07
65.	O	Hexadecane	1600	0.26	±0.01	0.30	±0.05	0.32	±0.01	0.29	±0.01	0.36	±0.01
66.	OS	Humuleneepoxide	1605	0.21	±0.00	0.22	±0.04	0.30	±0.01	0.28	±0.01	0.26	±0.01
67.	OS	Humuleneepoxide II	1613	0.65	±0.02	0.69	±0.06	0.87	±0.01	0.89	±0.04	0.80	±0.01
68.	OS	epi-Cubenol	1619	1.55	±0.04	1.73	±0.12	1.70	±0.04	1.77	±0.08	2.01	±0.01
69.	OS	γ-Eudesmol	1628	0.25	±0.01	0.28	±0.04	0.32	±0.03	0.29	±0.01	0.20	±0.02
70.	OS	Isospathulenol	1638			0.17	±0.04	0.24	±0.02	0.25	±0.03	0.18	±0.01
71.	OS	Caryophylla-4(12),8(13)-dien-5β-ol	1642	0.30	±0.00	0.47	±0.04	0.26	±0.37	0.62	±0.12		
72.	OS	τ-Cadinol	1648	12.96	±0.69	14.35	±0.64	13.65	±0.14	14.17	±0.69	16.63	±0.06
73.	OS	α-Muurolol	1655	0.44	±0.02	0.47	±0.06	0.54	±0.01	0.53	±0.00	0.59	±0.00
74.	OS	α-Eudesmol	1659	0.46	±0.03	0.51	±0.06	0.55	±0.01	0.53	±0.07	0.69	±0.01
75.	OS	α-Cadinol	1662	1.33	±0.04	1.13	±0.06	1.35	±0.06	1.36	±0.02	1.08	±0.04
76.	OS	9-Cedranone	1667	1.13	±0.04	1.29	±0.16	1.33	±0.01	1.37	±0.06	1.41	±0.07
77.	O	Cadalene	1675	1.87	±0.05	1.58	±0.16	2.34	±0.04	2.53	±0.18	2.03	±0.01
78.	OS	α-Bisabolol	1681	0.77	±0.04	0.87	±0.09	0.96	±0.03	0.92	±0.04	0.91	±0.00
79.	OS	epi-α-Bisabolol	1691	0.69	±0.03					0.38	±0.53		
80.	OS	*cis*-14-nor-Muurol-5-en-4-one	1693	2.68	±0.08	3.72	±0.22	3.72	±0.05	3.37	±0.7	4.45	±0.03
81.	O	Heptadecane	1703			0.28	±0.05	0.30	±0.01	0.13	±0.18	0.31	±0.00
82.	O	5-Ethyl-5-methylpentadecane	1709	0.27	±0.01	0.31	±0.06	0.38	±0.01	0.33	±0.03	0.41	±0.01
83.	O	Pentadecanal	1714	0.46	±0.04	0.53	±0.07	0.57	±0.01	0.52	±0.01	0.68	±0.01
84.	OS	(*E*,*E*)-Farnesol	1720	1.20	±0.07	1.43	±0.14	1.32	±0.04	1.45	±0.04	1.57	±0.02
85.	O	5-Phenyldodecane	1733	0.50	±0.05	0.55	±0.13	0.68	±0.02	0.62	±0.03	0.81	±0.01
86.	OS	Bisabolol oxide A	1750	1.97	±0.12	2.26	±0.27	2.23	±0.03	2.13	±0.12	2.60	±0.03
87.	OS	(*E*)-α-Atlantone	1777	0.12	±0.02	0.19	±0.03	0.19	±0.00	0.22	±0.03	0.23	±0.05
88.	O	Octadecane	1805			0.13	±0.02	0.20	±0.03			0.27	±0.02
89.	DT	Phytane	1811									0.21	±0.00
90.	O	Diisobutylphthalate	1872					0.24	±0.01	0.25	±0.08	0.33	±0.01
91.	DT	m-Camphorene	1957					0.14	±0.01			0.26	±0.02
92.	O	Eicosane	2003									0.12	±0.08
93.	O	Octadecanal	2021									0.20	±0.03
94.	O	1-Octadecanol	2088					0.29	±0.03			0.66	±0.08
95.	O	1-Tricosene	2296									0.31	±0.16
96.	O	Tricosane	2300							0.25	±0.13	0.44	±0.09
97.	O	2-Heneicosanone	2307					0.66	±0.20	0.84	±0.21	2.22	±0.35
		Total identified [No.]		82		81		81		83		83	
		Total identified [%]		99.29		99.95		99.95		99.69		99.72	
		Monoterpene hydrocarbnons (MH)		10.40		7.90		7.98		7.59		5.57	
		Oxygenated monoterpenes (OM)		37.19		38.23		31.34		27.77		28.65	
		Sesquiterpene hydrocarbons (SH)		11.22		10.27		12.36		14.35		12.06	
		Oxygenated sesquiterpenes (OS)		36.34		38.96		41.21		43.36		43.06	
		Diterpenes (DT)		-		-		0.14		-		0.47	
		Other (O)		4.14		4.59		6.92		6.62		9.91	

**RI:** Retention indices relative to n-alkanes (C_7_-C_40_) on HP-5MS capillary column; -: Not detected.

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
