# Peer review of "Gas Chromatography-Mass Spectrometry (GC-MS) Analysis of Essential Oils from AgNPs and AuNPs Elicited Lavandula angustifolia In Vitro Cultures"

_molecules, 2019, doi:10.3390/molecules24030606_

Round 1
Reviewer 1 Report
The submitted manuscript include interesting results. However some modifications should be reported to the text as I specified in details in my revision. See the attache pdf file.

Author Response
Reviewer 1
We appreciate the time and efforts of the editor and referee in reviewing this manuscript. We appreciate the reviewer’s insightful comments. We have addressed all issues indicated in the review report, and believe that the revised version meets the journal publication requirements.
We included the literature suggested by the Reviewer. Like the Reviewer, we believe that it will be necessary to carry out tests to determine the toxicity of nanoparticles. We have included such a conclusion in the summary.
Reviewer 2 Report
The Authors, proposed a chromatographic analysis of secondary metabolites from elicited Lavandula angustifolia cultures. However, the paper focuses on evaluation of addition effect of nanoparticles on the production of secondary metabolites (medicinal compounds). Therefore, the manuscript title for more in line with the research topic should be changed.
The article can be interesting for readers and hence, I have recommended publication following revisions. Below are first some more general comments on the article followed by some more specific comments. I hope the author considers incorporating these into their revision as I strongly believe that these will improve the quality of the article.
Did the nanoparticles have no toxic effect and did not slow down the plant growth? Have there been any anatomical or morphological changes in the plants? How the concentration of nanoparticles in the medium was selected? 50 mg is quite high level.
Did the authors investigate the toxic effects of nanoparticles? For example, a mutagenic effect using the RAPID method?
The authors should explain each abbreviation used in the text. Before using the symbol, it should be expanded.
Please take into account the addition of short paragraph with physicochemical properties of the most important essential oils. Because, their content is dependent on the physicochemical reactions and properties of nanoparticles.
Line 20 should be improved on: The addition of AuNPs and AgNPs to the media resulted in decreasing of lower molecular weight compounds.
Line 35 A reference to this statement should be added.
line 46: dot is missing.
line 52 This sentence should be corrected.
line 62 The Authors should explained the limitation of using gold and silver nanocolloids as elicitors.
lines 67-68 The Authors stated, that there have been no literature reports so far regarding the influence of nanoparticles on the production of valuable secondary metabolites, such as essential oils, in plants propagated in in vitro cultures. That is not true. Please explain the novelty of proposed research. If not, please specify in more detail the scope of the proposed research.
Please see related papers: https://doi.org/10.1016/j.ijpharm.2017.01.013 ; https://doi.org/10.1016/j.ecoenv.2018.10.017 ; https://doi.org/10.1016/j.cis.2016.04.008
line 81 dot is missing.
line 84 What the retention indices means? Is not should be retention indexes?
What mean the symbols (a, b and c) presented in Table 2?
Table 3 How many repetition of percentage calculations were performed? How was possible that standard deviations in some cases were 0?
The Authors stated that the mechanism of nanoparticles impact on plant cells has not been thoroughly examined yet. Then compared the results obtained by another Authors from literature. Thus, the Authors deny themselves that this is the first work on this subject.
Moreover, still is lack the proposition of mechanism of action.
The manuscript should be "spellchecked" and "grammar-checked" carefully. There are language and punctuation errors.
The units used should be unified. The concentrations are once in mM and once in mg dm-3 units.
According to chromatographic analysis: How was the quantitative analysis was performed? How was the retention indexes calculated? Why the temperature programme was so long? If the Authors will check the boiling points of analytes, the shorter programme can be proposed.
Why the Authors operated the MS in SCAN mode? How are the concentration levels of essential oils in real samples? The injected concentration used by Authors was 0.01 mg. Is this correct?

Author Response
Reviewer 2
We appreciate the time and efforts of the editor and referee in reviewing this manuscript. We appreciate the reviewer’s insightful comments. We have addressed all issues indicated in the review report, and believe that the revised version meets the journal publication requirements.
The Authors, proposed a chromatographic analysis of secondary metabolites from elicited Lavandula angustifolia cultures. However, the paper focuses on evaluation of addition effect of nanoparticles on the production of secondary metabolites (medicinal compounds). Therefore, the manuscript title for more in line with the research topic should be changed.
Thank you for this attention. Initially, the article was to be published under the title “AgNPs and AuNPs nanocolloids change the composition of essential oils of Lavandula angustifolia Mill. propagated in in vitro cultures”, but due to the fact that it will be included in the special edition "Technology for Natural Products Research" we changed the title to the current one, to emphasize the role of “technology”.
I leave the editor the decision which one will be better.
Did the nanoparticles have no toxic effect and did not slow down the plant growth? Have there been any anatomical or morphological changes in the plants? How the concentration of nanoparticles in the medium was selected? 50 mg is quite high level.
The concentration of nanoparticles in the medium was determined after previous studies, which investigated the effect of nanoparticles on plant development. These studies are due to be published in another article.
The concentration of 50 mg·dm-3 AuNPs and AgNPs decreased plant growth, but it was not high enough that the effect would be lethal. The multiplication of lavender on medium, even with the highest concentration of AuNPs and AgNPs used, allowed us to obtain a large amount of horse biomass for the isolation of essential oils.
Did the authors investigate the toxic effects of nanoparticles? For example, a mutagenic effect using the RAPID method?
This is a very interesting question that we may be looking to answer in the future.
We are aware of publications that indicate the toxic effect of nanoparticles, including those indicating a mutagenic effect. In our research, we have not studied the effect of nanoparticles on the plant genotype. The goal of our study was to obtain a large amount of biomass, which will be destroyed during the process of isolation of secondary metabolites. In this case potential mutations will not be inherited for future generations.
The authors should explain each abbreviation used in the text. Before using the symbol, it should be expanded.
Thank you for your comment, the abbreviations have been explained and corrected in the work.
Please take into account the addition of short paragraph with physicochemical properties of the most important essential oils. Because, their content is dependent on the physicochemical reactions and properties of nanoparticles. ?????
We have added a paragraph on the properties of essential oils to the introduction chapter. Physicochemical properties are also discussed in the results and discussion chapter.
Line 20 should be improved on: The addition of AuNPs and AgNPs to the media resulted in decreasing of lower molecular weight compounds.
Thank you for your comment, I have made corrections.
Line 35 A reference to this statement should be added.
Thank you for your comment, I have made corrections.
Line 46 and 81: dot is missing.
Thank you for your comment, I have made corrections.
Line 52 This sentence should be corrected.
Thank you for your comment, I have made corrections. This sentence now reads as follows: „ Ongoing studies are attempting to determine the suitability of nanoparticles as elicitors in the in vitro cultures.”
Line 62 The Authors should explained the limitation of using gold and silver nanocolloids as elicitors.
The sentence previously read: So far, the attempts to use gold and silver nanocolloids as elicitors in plant in vitro cultures have been limited. It has beed changed to the following: The number of research papers in which gold and silver nanoparticles were used as elicitors in in vitro culture was low.”, which better reflects the intentions of the authors.
Lines 67-68 The Authors stated, that there have been no literature reports so far regarding the influence of nanoparticles on the production of valuable secondary metabolites, such as essential oils, in plants propagated in in vitro cultures. That is not true. Please explain the novelty of proposed research. If not, please specify in more detail the scope of the proposed research.
Please see related papers: https://doi.org/10.1016/j.ijpharm.2017.01.013;
https://doi.org/10.1016/j.ecoenv.2018.10.017;
https://doi.org/10.1016/j.cis.2016.04.008
As we have previously stated, there are no existing literature reports regarding to the influence of nanoparticles on the production of essential oils by plants propagated in in vitro cultures.
The above references suggested by the reviewer, concern the reverse process, the so-called green synthesis of nanoparticles, by plants in in vitro cultures, using essential oils.
What mean the symbols (a, b and c) presented in Table 2?
Thank you for your comment, I have made corrections.
Table 3: a, b, c - Values followed by the same letter are not significantly different at P ≤ 0.05 according to the LSD Tukey’s test.
Table 3 How many repetition of percentage calculations were performed? How was possible that standard deviations in some cases were 0?
Three repetitions of percentage calculations were performed.
Under table 3 we have added: ± standard deviation (n = 3)
Moreover, the section titled “Essential oil extraction” has been modified: Fifteen grams of the entire dried aerial parts of lavender were placed in 1000 ml round-bottomed flasks along with 400 ml of distilled water. They were then subjected to hydrodistillation (3 replicates) for two hours using a Clevenger apparatus according to the method recommended by the European Pharmacopoeia 5.0 [43].
The Authors stated that the mechanism of nanoparticles impact on plant cells has not been thoroughly examined yet. Then compared the results obtained by another Authors from literature. Thus, the Authors deny themselves that this is the first work on this subject.
The above statements were in fact untrue, and the paragraph has been corrected. The authors wanted to emphasize that this is the first work on the influence of nanoparticles on the amount and composition of essential oils produced by plants in in vitro cultures.
The manuscript should be "spellchecked" and "grammar-checked" carefully. There are language and punctuation errors.
Thank you for your comment, I have made corrections.
The units used should be unified. The concentrations are oncein mM and once in mg·dm-3 units.
Thank you for your comment, I have made corrections.
According to chromatographic analysis:
· What the retention indices means? Is not should be retention indexes? How was the retention indexes calculated?
The term retention indices (RI) is used interchangeably with the term retention indexes.
The retention indices (RI) were calculated for all constituents using a homologus series of n-alkanes (C7-C40) injected under the same chromatographic conditions as the analyzed oil samples. References 46-47.
46. Van Den Dool, H.; Kratz, P.D. A generalization of the retention index system including linear temperature programmed gas-liquid partition chromatography. J.Chromatogr. A. 1963, 11, 463-471.
47. Babushok, V.I..; Linstrom, P.J.; Zenkevich, I.G. Retention indices for frequently reported compounds of plant essential oils. J. Phys.Chem. Ref. Data, 2011, 40, 043101-42.
· How was the quantitative analysis was performed?
· Why the temperature programme was so long? If the Authors will check the boiling points of analytes, the shorter programme can be proposed.
Relative percentage amounts of the lavender oil components were evaluated from the total peak area (TIC) by apparatus software. The same method of quantification of essential oil constituents has been applied by Hassanpouragdam et al. (reference 44) and Rosas et al. (reference 45).
44. Hassanpouraghdam, M.B.; Hassani, A.; Shalamzari M.S. Menthone – and estragole – rich essential oil of cultivated Ocimum basilicum L. from Northwest Iran. Chemija 2010, 21, 59-62.
45. Rosas, J.F.; Zoghbi, M.G.B.; Andrade, E.H.A.; van den Berg, M.E. Chemical composition of a methyl-(E)-cinnamate Ocimum micranthum Willd. from the Amazon. Flavour Fragr. J. 2005, 20, 161-163.
Chromatography is, generally, a slow method of analysis – analysis time often exceeds one hour when a complex mixture must be separated. Reduction of the retention times of separated components requires the use of a short column containing a thin (0,1μm) film of stationary phase. This column must operate with a rapid temperature gradient (e.g. 100°C/min).
We used such a long temperature programme in order to be sure that all analytes has been eluted from the column.
· Why the Authors operated the MS in SCAN mode? How are the concentration levels of essential oils in real samples?
Scan mode is used for the identification of chemical components using a mass spectrum.
We identified essential oil constituents by comparison of their mass spectra with those stored in the Wiley NBS75K.L and NIST/EPA/NIH (2002 version) mass spectral libraries using various search engines (PBM, Nist02).
· The injected concentration used by Authors was 0.01 mg. Is this correct?
We don’t know how the reviewer calculated this concentration (0,01 mg) of essential oil.
We suspect that our original specification of the method by which we prepared samples for GC-MS analysis was incomprehensible.’. We have corrected this section of the manuscript.
We would like to thank you for all corrections, which have undoubtedly contributed to the improvement of the manuscript.